# Partial Biodegradable Blend for Fused Filament Fabrication: In-Process Thermal and Post-Printing Moisture Resistance

**DOI:** 10.3390/polym14081527

**Published:** 2022-04-09

**Authors:** Muhammad Harris, Hammad Mohsin, Rakhshanda Naveed, Johan Potgieter, Kashif Ishfaq, Sudip Ray, Marie-Joo Le Guen, Richard Archer, Khalid Mahmood Arif

**Affiliations:** 1Massey Agrifood Digital Lab, Massey University, Palmerston North 4410, New Zealand; j.potgieter@massey.ac.nz; 2Industrial and Manufacturing Engineering Department, Rachna College of Engineering and Technology, Gujranwala 52250, Pakistan; 3Department of Polymer Engineering, National Textile University, Faisalabad 37610, Pakistan; mhammad@ntu.edu.pk; 4Industrial and Manufacturing Engineering Department, University of Engineering and Technology, Lahore 54890, Pakistan; rakhshanda@uet.edu.pk (R.N.); kashif.ishfaq@uet.edu.pk (K.I.); 5New Zealand Institute for Minerals to Materials Research, Greymouth 7805, New Zealand; s.ray@auckland.ac.nz; 6Scion, Rotorua 3046, New Zealand; mariejoo.leguen@scionresearch.com; 7School of Food and Advanced Technology, Massey University, Palmerston North 4410, New Zealand; r.h.archer@massey.ac.nz; 8Department of Mechanical and Electrical Engineering, SF&AT, Massey University, Auckland 0632, New Zealand; k.arif@massey.ac.nz

**Keywords:** fused deposition modeling, polypropylene, polylactic acid, moisture-based degradation, pellet 3D printing

## Abstract

Despite the extensive research, the moisture-based degradation of the 3D-printed polypropylene and polylactic acid blend is not yet reported. This research is a part of study reported on partial biodegradable blends proposed for large-scale additive manufacturing applications. However, the previous work does not provide information about the stability of the proposed blend system against moisture-based degradation. Therefore, this research presents a combination of excessive physical interlocking and minimum chemical grafting in a partial biodegradable blend to achieve stability against in-process thermal and moisture-based degradation. In this regard, a blend of polylactic acid and polypropylene compatibilized with polyethylene graft maleic anhydride is presented for fused filament fabrication. The research implements, for the first time, an ANOVA for combined thermal and moisture-based degradation. The results are explained using thermochemical and microscopic techniques. Scanning electron microscopy is used for analyzing the printed blend. Fourier transform infrared spectroscopy has allowed studying the intermolecular interactions due to the partial blending and degradation mechanism. Differential scanning calorimetry analyzes the blending (physical interlocking or chemical grafting) and thermochemical effects of the degradation mechanism. The thermogravimetric analysis further validates the physical interlocking and chemical grafting. The novel concept of partial blending with excessive interlocking reports high mechanical stability against moisture-based degradation.

## 1. Introduction

Fused filament fabrication (FFF), also known as fused deposition modeling (FDM), is prominent among all additive manufacturing (AM) techniques due to various reasons, such as low cost, easy availability, and simple process [1,2,3]. Large-scale additive manufacturing using FFF is one of the highly rated research developments in this decade [4,5,6]. In this regard, big area additive manufacturing (BAAM) is the earliest experimental FFF setup [7]. The BAAM employs acrylonitrile butadiene styrene (ABS) with carbon fibers as one of the first large-scale FFF materials [8]. The inclusion of carbon fibers is attributed to the high mechanical strength of reinforced carbon fibers [8,9]. However, the reported materials for large-scale structures are not yet investigated in real (humid or moisture-based) environments.

Polylactic acid (PLA) is the only biodegradable polymer in FFF that proves its commercial feasibility in the light of numerous reasons [10]. In comparison to ABS, the PLA provides superior mechanical properties with the additional benefit of biodegradability [10]. Therefore, PLA qualifies the requirements to be considered as a feasible polymer for large-scale additive manufacturing [11]. However, the real-time testing of PLA in severe environments exposes the high vulnerability to chemical chain scission. This is owing to the poor moisture-based stability of neat PLA [12,13]. The literature reports scission or depletion of C-O-C and C=O in FTIR analysis [13]. The weak intermolecular chains can cause catastrophic failure of the large-scale structures in severe environments (moisture or humid). Therefore, PLA requires particular attention for making the intermolecular chains strong enough to withstand moisture-based degradation.

Among all researched approaches, the simplest one to overcome chain scission is the polymer blending with high-temperature, fossil fuel-based polymers [14,15]. For example, nylon is blended with PLA to gain good mechanical properties. Similarly, ABS blended with PLA reports good thermomechanical properties. The limitation to the abovementioned reported blends is the high percentage of non-biodegradable polymer, usually greater than 25% [13,14,15], which raises serious questions regarding the harmful effects on the natural eco-system. Furthermore, the aforementioned blends are not tested for real environmental degradation, such as the moisture-based degradation [13].

One of the potential non-biodegradable polymers for improving the thermomechanical stability of PLA is polypropylene (PP). Long et al. [16] reports a natural fiber (Bamboo)-reinforced PP and PLA blend system. The system is also compatibilized with polypropylene graft maleic anhydride (PP-g MAH). The blend system provides a maximum of 28.1 MPa for a blend system with 52.5% PP and just 22.5% PLA [16]. The overall blend system was not eco-friendly with such a high percentage (52.5%) of non-biodegradable PP. It is also noted that the approach of achieving compatibilization through PP-g MAH does not provide sufficient mechanical properties [16]. Furthermore, the hydrophilic nature of bamboo fibers can potentially cause moisture-based degradation to PLA that is not investigated in the proposed PLA/PP/PP-g-MAH/fiber blend. The moisture-based degradation can potentially damage the intermolecular chains of PLA, which will cause detrimental damages in case of large-scale applications. Therefore, the research needs a proper approach for making a strong intermolecular structure of the PLA/PP blend that can withstand enzymatic biodegradation and moisture-based degradation.

It is important to note that the blend system of PP and PLA is extensively reported for non-3D printing applications. For the sake of simplicity, the literature associated with PLA and PP can be divided into different categories. These include compatibilized blends [17], compatibilized blends with modification agents (toughening additives) [18], non-compatibilized blends [19], ternary blends, and fiber-added blends. None of the abovementioned categories include 3D printing along with moisture-based degradation analysis, as shown in Table 1. Furthermore, none of the reported compositions of PP with PLA have been tested for the statistical design of experiments to ensure the properties in the blend.

This research is the continuation of a series of projects on evaluating the potential of different chemical approaches (compatibilization, or physical interlocking, or both) to achieve optimal stability against real-time environmental degradation mechanisms (soil, moisture, and thermal) [13]. In this regard, the physical interlocking is noted with a significant contribution to achieve high stability against degradation mechanisms [13]. However, the previous experiments include the physical interlocking combined with sufficient chemical grafting using a suitable compatibilizer [13]. The recent research reported by the authors of this study develops a blend of polypropylene (PP) and high-density polyethylene (HDPE) with excessive physical interlocking and minimum chemical grafting. The novel approach allows to achieve good thermal resistance [20]. However, the stability of the proposed blend system is not reported against moisture-based degradation [20], which can be a detrimental degradation phenomenon for PLA-based blends.

**Table 1 polymers-14-01527-t001:** Literature review for PP and PLA blend system.

Blend	Minimum Percentage of PP	Properties	3D Printing	Moisture-Based Degradation of 3D-Printed Blend
Non-compatibilized PP and PLA [19]	30	Viscosity Non-Newtonian index Tensile stress and strain	No	No
Compatibilized PP and PLA PP-g-MAH SEBS-g-MAH [17]	80	Tensile strength Impact strength Morphology Complex viscosity	No	No
Non-compatibilized PP and PLA fibers (not bulk) [21]	20	Breaking tenacity AFM-based morphology X-ray diffraction graphs	No	No
Compatibilized rPP, PLA, n-(6-aminohexyl) aminomethyltriethoxysilane [22]	10	Phase morphology TGA, DSC, Impact strength Tensile strength	No	No
Compatibilized PP, PLA, Toughening modifier Hybrid compatibilizer of following PP-g-MAH PE-g-GMA [18]	60	Tensile strength Flexural strength Impact strength Viscosity Relaxation time analysis	No	No
Compatibilized PP, PLA, EBA-GMA [23]	10	TGA, DSC, SEM Viscosity	No	No
Compatibilized PLA, PP, PP-g-MAH [24]	64		No	No
Compatibilized PLA, PP, PP-g-MAH [25]	80	TGA Viscosity Thermal degradation activation energy	No	No
Compatibilized PP, PLA, PP-g-MAH, OMMT [26]	25	TGA SEM Impact testing Tensile testing	No	No

This research work proposes the excessive physical interlocking and partial chemical grafting of PLA with minimum PP to overcome the moisture-based degradation. The physical interlocking is further increased through the introduction of a partial compatibilizer (high-density polyethylene graft maleic anhydride, HDPE-g-MAH) in the proposed blend system (PLA/PP). Based on the literature, the PLA will probably show good compatibilization with maleic anhydride (MAH) and physical interlocking with PP [14]. The overall excessive, physically interlocked PP and HDPE will probably provide good mechanical properties after moisture degradation for FFF structures. This research also reports statistical ANOVA to analyze the degradation effects on the PLA/PP/HDPE-g-MAH blend system. The aspects regarding polymer chemistry are thoroughly discussed using scanning electron microscopy (SEM), Fourier transform infrared spectroscopy (FTIR), differential scanning calorimetry (DSC), and thermogravimetric analysis (TGA).

## 2. Materials and Methods

### 2.1. Materials

Neat PLA (NatureWorks^®^ Ingeo™ 2002D) was purchased from SCION, Rotorua, New Zealand. The specific weight of neat PLA was 1.24 g/cm^3^. HDPE-g-MAH (95:5 by weight %) was provided by Shenzhen Jindaquan Technology Co. Ltd., Shenzhen, China. PP (Moplen HP400N) was purchased from TCL Hunt, Auckland, New Zealand. The specific weight of PP was 0.905 g/cm^3^ alongside a melt flow index (MFI) of 11 g/10 min.

### 2.2. Melt Blending (Until Successful 3D Printing)

A thermostat blast oven from HST, China, was used to dry all three polymers for 1 h at 40% up until 10 min. Single-screw extrusion (HAAKE™ Rheomex OS) was used to perform the melt blending at Scion, Rotorua, New Zealand. The extruded filament from the single-screw extruder was pelletized into cylindrical pellets of 1.5 ± 0.5 mm. Single-screw extrusion was preferred over twin-screw extrusion to avoid thermomechanical degradation due to the excessive shear in the twin-screw setup [27,28]. The single-screw extrusion helped to achieve the properties of the blend near to a blend made without any significant degradation [27,28] of PLA and PP.

This research was specifically intended for the development of 3D printing material for FFF with minimum non-biodegradable polymer. Therefore, “3D printing” was set as the main criterium for finalizing the compositions of the novel blend system. Each blend composition was prepared in a single-screw extrusion followed by 3D printing on the FFF machine. A successive number of blend compositions were prepared until the desired 3D printing sample was obtained. In this regard, the first composition was prepared as per the desired objective of this research, i.e., minimum PP and HDPE-g-MAH contents. Based on [29,30,31,32,33], 20% PP and 5% HDPE-g-MAH were blended with 75% neat PLA [31,32,33]. However, the first composition resulted in large die swelling during 3D printing. The abnormal die swelling was caused by the large weight percent of MAH, as per [13].

Based on the undesired rheological effects of large compositions, the next blend composition was managed with less PP and HDPE-g-MAH, i.e., 7.5% PP and 0.5% HDPE-g-MAH [13,33]. The second composition resulted in no die swelling, and the obtained filament had dimensions of 0.2 ± 0.1 mm. Therefore, a subsequent composition was not prepared. The blend compositions are provided are in Table 2.

### 2.3. Pellet 3D Printing

Fused deposition modeling (FDM) was performed with a 3D pellet printer, custom built at the School of Advanced Manufacturing and Technology, Massey University, Palmerston North, New Zealand, as shown in Figure 1 [34]. The literature reported significant intermolecular degradation due to the combined thermal and mechanical shearing in the extruder [27,34,35], which was not desired in this research. Therefore, four key modifications were made in the pellet printer to print the novel blend material as near as possible to the as-prepared blend (Figure 1). For example, (1) Teflon insulative plate, (2) larger milled slot of cooling jacket, (3) SLS cone for hopper, and (4) variable length extruder. First, the Teflon plate was used as an insulative barrier to stop the propagation of temperature from the lower barrel of the extruder to the hopper (upper part). Second, the liquid cooling system was improved with the help of a larger milled slot to carry a higher volume flow rate of the coolant. The larger volume flow rate achieved better dissipation of the barrel heat. Third, the SLS-printed polymeric cone aimed to avoid the pre-heating of the fed pellets before 3D printing. The SLS cone maintained the temperature inside the hopper by providing an insulation between the aluminum hopper and the fed pellets. Fourth, the variable length extruder mechanism was used to increase the gap between the nose of the lead screw and the surface of the barrel (≈15 mm). The increased gap avoided the thermomechanical shearing of the melt blend. Therefore, the four abovementioned modifications were aimed to 3D print the blend with minimum thermomechanical deteriorations, as near as possible to the properties of the as-prepared blend.

The pellet printer is operated with “Pronterface” software. “Pronterface” executed the G-codes for the sliced files in “Stl” format of ASTM D638 type IV dog bone. For performing the slicing, software named “Slic3r” was utilized, which also developed the G-codes. The printing parameters are provided in Table 3.

### 2.4. Water Absorption (Moisture) Testing

Water absorption testing was performed for the analysis of moisture-based degradation. ASTM D638 type IV dog bones [36] were printed of neat PLA and the novel blend. The dog bones were immersed in water for 45 days in air-tight plastic containers as per [13]. The water was properly changed at scheduled intervals. The weight of each sample was measured before immersion (m_0_). After 45 days, the samples were taken out of water, dried with paper towels, and acclimatized at 25 ± 3 °C. Then, each acclimatized sample was weighed to obtain m_1_. Mass gain (%m_G_) in percentage was calculated using the following relation [37]:(1)$${\%m}_{G}=\frac{m_{1}-m_{0}}{m_{0}}\times 100$$

The water absorption analysis was designed with “multiple leveled, general full factorial ANOVA”. The stability of the FFF structure was interpreted in terms of adhesion between the extruded beads, which was dependent upon the bed and printing temperature [12]. Therefore, it was aimed to analyze the effects of water absorption (moisture-based degradation) on different levels of adhesion achieved with variable bed and printing temperatures.

In this regard, a range of trial experiments were performed to find the minimum and maximum limits of printing temperatures. The trial revealed no 3D printing due to the clogging of the polymer blend inside the pellet printer heating barrel below 157 °C. The clogging was caused due to the printing (barrel) temperature being lower than the melting point of the polymer. On the other hand, the temperature above 174 °C caused severe degradation (burning) of the polymer blend inside the printing nozzle. Therefore, the printing temperature was set between 161 and 171 °C.

The bed temperature was also selected based on the printability of the polymer blend. The lowest limit for the bed temperature was decided as the minimum temperature (25 °C) of the 3D pellet printer. For the upper limit of the bed temperature, the printed material was noted with softening above 90 °C. The softening affects the overall thickness of the printed sample. Therefore, the bed temperature was selected in the range of 25 to 85 °C.

The factors and levels are provided in Table 4.

### 2.5. In-Process (During 3D Printing) Thermal Testing

ASTM D638 Type IV dog bones were selected for tensile testing of samples printed with variations in in-process thermal variables (printing temperature and bed temperature). The average of tensile strength was calculated to present the comparison of effects of bed temperature and printing temperature. The reasons for considering the in-process variables are described below.

The thermal properties of any material can be significantly modified based on differences in processing temperatures. The reason for such modifications is reported due to the change in crystallographic regions or crystallographic orientations. As polymers have amorphous regions, therefore, the impact can also be the degradation of amorphous regions due to the increase of melting temperatures during processing. A similar case is implemented in 3D printing, where the change in in-process thermal variables can cause significant variations in overall intermolecular interactions. The two most important thermal variables are printing temperature and bed temperature [38]. The literature reports a significant impact of printing and bed temperatures on overall polymeric structures that causes changes in mechanical properties [38].

### 2.6. Tensile Testing

The tensile testing was performed on Instron 5967, Norwood, MA, USA. The machine uses a load cell (30 KN) and an extensometer. The minimum gauge span of the extensometer was 25 mm. The tensile testing was performed at an extension rate of 5 mm/min. The average of multiple samples’ tensile strength was considered for ANOVA.

ASTM D638 type IV dog bones were 3D-printed with an overall length of 115 mm, gauge length of 25 mm, distance between gap of 65 mm, and thickness of 4 mm [36].

### 2.7. Scanning Electron Microscopy

SEM analysis was performed on a Hitachi TM3030 (Hitachi, Tokyo, Japan). SEM was used to analyze the nature of melt blending, i.e., phase separation or physical interlocking.

### 2.8. Fourier Transform Infrared Spectroscopy

FTIR was performed on a Thermo electron Nicolet 8700 spectrometer, Thermofisher Waltham, MA, USA. An average of 30 spectra was considered to measure the FTIR transmittance in the range of 400–4000 cm^−1^. FTIR analyzed the effects on intermolecular interactions after moisture treatments. Furthermore, the modifications in the intermolecular chains after blending were also observed in the FTIR analysis to confirm the effectivity of the pellet printer. The machine used the OMNIC E.S.P software (version 7.1) to perform the normalization and correction of each spectrum with respect to the software-generated baseline.

### 2.9. Differential Scanning Calorimetry

DSC analysis was carried out on a NETZSCH simultaneous thermal analyzer 449 F1 Jupiter, Selb, Germany. The machine was operated in a temperature range of 25 to 550 °C. The temperature was increased at a rate of 10 °C/min at 50 mL/min of nitrogen purging. DSC was used to analyze two aspects: (1) the nature of the chemical interaction (grafting or physical interlocking), and (2) effects of moisture-based degradation on thermal properties.

### 2.10. Thermogravimetric Analysis

TGA was carried out on a simultaneous thermal analyzer 449 F1 Jupiter from NETZSCH, Selb, Germany. The machine was operated in a temperature range of 25 to 550 °C, increased at a rate of 10 °C/min. The nitrogen purging (50 mL/min) was also used to perform the tests. TGA was used to quantitatively evaluate the following aspects: physical interlocking and stability of the printed blend after moisture treatments.

## 3. Results

### 3.1. Water Absorption (Moisture-Based Degradation)

The effects of water absorption on mass gain for the neat PLA and the blend are shown in Figure 2. Overall, the mass gain percentage decreased with an increase of the bed temperature and printing temperature for the blend. This decrease of mass gain percentage can be attributed to the enhanced adhesion (fusion) between beads due to the increase of temperatures (bed and printing) [12]. Furthermore, the maximum mass gain for PLA/PP/PE-g-MAH (0.39%) was far less than neat PLA. The comparative low mass gain showed excellent resistance to water absorption of the novel blend as compared to neat PLA.

### 3.2. Tensile Testing

The result for tensile testing is presented in Figure 3 The ANOVA for tensile testing of water absorbed samples was found with an outlier (Figure 3b) that resulted as insignificant for all variables (Figure 3a). The outlier (79 MPa) was associated with the treated highest temperature combination (171 °C, 85 °C). Therefore, the ANOVA needs to be repeated for new samples (171 °C, 85 °C). The repeated ANOVA provided printing temperature and moisture treatment as significant variables (Figure 3c). However, the moisture treatment was found with a strange increase of tensile strength, instead of a decrease (Figure 3d). The reasons for this strange behavior are investigated in the Discussion Section. The detailed ANOVA for moisture treatment is also provided in Appendix A. The design of experiments (DoE) for water absorption testing is provided in Table 5.

### 3.3. Effects of Bed and Printing Temperatures

The effects of in-process printing temperature and bed temperature on tensile strength are shown in Figure 4.

For effects of printing temperatures at a constant bed temperature, the highest of 44.9 MPa was obtained for 171 °C at 25 °C and for 161 °C at 85 °C. At a bed temperature of 85 °C, the tensile strengths at the three printing temperatures were among the highest and also nearly similar.

For effects of the bed temperature at constant printing temperatures, the highest of 44.9 MPa was obtained for 85 °C at a printing temperature of 161 °C. At a printing temperature of 171 °C, the tensile strength was found to be similar for 25 °C (43.4 MPa) and 85 °C (43.1 MPa).

Overall, it is noted that the change in in-process thermal variables does affect the overall properties. However, at the constant high-temperature combinations of 85 °C (bed temperature) and 171 °C (printing temperature), the properties were revealed to be similar.

## 4. Discussion

### 4.1. Microscopic Analysis of Melt Blending

As achievement of excess physical interlocking in melt blending is the main goal in this research, therefore, the 3D-printed blend was tested for visual signs of interlocking. Furthermore, the after-effects of tensile loading on blend morphology were also observed.

The images in Figure 5 reveal visible signs of polypropylene interlocked in the PLA matrix. The figure shows magnified images of the area loaded under tensile loading. The 4000-times magnified image presents two distinct morphologies. One of them is fibrous and the other is in stable beads. The fibers are probably PP as they appear distinct and pulled out from the stable beads. Similar kinds of distinct PP inclusions were also noted by Codou et al. [39].

The overall external morphology appeared random as a result of tensile testing. However, most of the internal bead structure appeared stable, with no fibrous fracture, as shown in Figure 5.

The morphology showed signs of physical interlocking; however, it requires chemical and thermochemical analysis to be further confirmed.

### 4.2. Analysis for Intermolecular Interactions

In the following discussion, the FTIR analysis was used to detect and analyze different aspects, such as blending, 3D printing, and moisture-based degradation. In this regard, the effects on chemical groups were recorded and interpreted using appropriate literature.

The FTIR spectra for neat polymers and compatibilizers are analyzed in Figure 6 and Table 6. In the neat PLA FTIR spectrum, C-O-C, C=O, and C-H [40,41] were observed at 1086, 1748, and 2997–2849 cm^−1^, respectively. In the PP FTIR spectrum, CH_2_ and CH_3_ [28,42] were detected in the range of 2800–3000 cm^−1^. In the PE-g-MAH FTIR spectrum, the C-H groups associated with HDPE were observed at 2917 and 2849 cm^−1^. Additionally, the C=O groups associated with MAH were detected at 1705 and 718 cm^−1^. The PE-g-MAH spectra correspond to the pertinent literature [13].

FTIR spectra of the blend printed at the low-temperature combination and the high-temperature combination were compared with the neat PLA (Figure 7). The melt blending reports the intermolecular interactions in different forms.

The first sign of intermolecular interactions was noted in the form of a wavelength-shift of various chemical groups (Figure 7). In neat PLA, the wavelengths of C-H [13] (2997 cm^−1^), C-O-C groups [11] (1085.0 to 1082.2 cm^−1^), and C=O groups [18] (1747 cm^−1^) were shifted to 2989, 1083, and 1743 cm^−1^, respectively. The second sign of intermolecular interactions was observed in the form of variations in intensities of chemical groups (Figure 7). The intensity of C=O groups [26] in the non-treated blend (161 °C, 25 °C) at 1746 cm^−1^ was increased to 85% as compared to the neat PLA (90%). The increase in C=O intensity is a result of the synchronization between similar chemical groups (C=O) of PLA and MAH after melt blending. The synchronization was detected in the FTIR spectrum of the non-treated blend in the form of a hump at 1705 cm^−1^. This hump is associated with the C=O of MAH, as found in the PE-g-MAH spectrum. The third sign of intermolecular interactions was found in the form of the appearance of a new peak as compared to PLA (Figure 7). In this regard, a fourth peak at 2950.0 cm^−1^ in the non-treated blend (161 °C, 25 °C) associated with saturated hydrocarbons of polypropylene was included in three C-H peaks of PLA. However, this fourth peak attributed to PP in the blend appeared with reduced intensity, i.e., 98% in the blend as compared to 88% in PP. The fourth distinct peak shows the phase separation and the reduced intensity shows the restricted mobility of PP chains [43]. The abovementioned FTIR analysis confirms the chemical effects of melt blending.

The effect of water absorption (moisture-based degradation) on the blend was also analyzed with the comparison of water absorbed combinations at low (161 °C, 25 °C) and high (171 °C, 85 °C) temperatures with the non-treated combination at the corresponding temperatures, respectively (Figure 8 and Table 6). The comparison of water absorbed at a low temperature (161 °C, 25 °C) revealed a strange increase of 5.1% (85–90.1%) for C-O-C in transmittance after moisture-based degradation, showing the stability. On the other hand, the high temperature (171 °C, 85 °C) caused a decrease of 2.1% (90.1–88%) for C-O-C groups, showing comparative degradation, as expected [13], due to the high printing and bed temperatures during 3D printing. One of the probable reasons for the increase in C-O-C transmittance is the enhanced reorientation of PLA chains (C-O-C) in the low-temperature degraded blend. In this regard, the enhanced reorientation is only possible in case of either of the following: (1) an increase of temperature [13], or (2) a decrease of the physical interlocking component [14], or (3) an increase of crystalline regions through a decrease of amorphous regions in PLA [12]. The former two reasons are not plausible as: (a) the water absorption testing is performed at room temperature, and (b) the PP cannot degrade at the same rate [28] as PLA to cause a decrease in physical interlocking. However, the amorphous regions of PLA (>60%) are highly susceptible to moisture-based degradation, which may lead to an ease of mobility for the remaining C-O-C chains of PLA to reorient into new crystalline regions.

The effects on intermolecular interactions were clearly observed in FTIR. However, the nature of the ternary blend still requires the analysis of the thermal profiles of non-treated and treated blends. In this regard, the subsequent discussion includes the DSC analysis.

### 4.3. Analysis for Nature of Blending and Effects of Degradation Mechanisms

DSC analysis was used to analyze the nature of chemical interactions (grafting or interlocking or both) between PLA and PP after the blending and degradation mechanism. DSC analysis was also performed to find thermochemical reasoning for significant and insignificant variations in different variables found in the ANOVA.

The non-printed blend pellets were analyzed for the effects of polymer blending on thermal properties with respect to neat PLA in Figure 9. The analysis was performed with respect to the glass transition (T_g_ and H_G_) phase, (b) melt crystallization (T_M_) phase, and (c) degradation (ΔH_D_) phase. First, the blend pellets were found with a reduced glass transition temperature of 63 °C, which was noted as 65 °C for neat PLA (Figure 9). On the contrary, the enthalpy of glass transition (ΔH_G_) for the blend pellets was noted as ≈4 J/g, which was higher than neat PLA (0.03 J/g). The reduction of Tg presented early intermolecular reorientation at a low temperature, and the increased ΔH_G_ showed a comparative high crystallite formation due to the probable partial compatibilization [44]. Second, the melt crystallization of the non-printed blend appeared bimodal as compared to the unimodal thermal profile of neat PLA (Figure 9). The bimodal thermal profile is highlighted with a magnified view of the melt crystallization peak (Figure 9) for non-printed pellets. The melt crystallization temperature (T_M_) of the non-printed pellets of the blend was also increased to 155.5 °C as compared to neat PLA (153 °C). The bimodal presentation of the melt crystallization peak presents the phase separation (physical interlocking) of PP and PLA [29], and the increased T_M_ was caused due to the partial chemical grafting [13]. Third, the degradation DSC peak of non-printed blend pellets was found with decreased enthalpy (ΔH_D_), which occurred due to the phase separation of PP and PLA [45]. This proves that the desired partial compatibilization and physical interlocking were successfully prepared.

The moisture treatment on the ternary blend is observed in Figure 9. The high-temperature combinations were detected with high enthalpy of cold crystallization (ΔH_C_), melt crystallization (ΔH_M_), and degradation (ΔH_D_) as compared to low-temperature combinations. The values at high-temperature combinations for ΔH_C_, ΔH_M_, and ΔH_D_ were 13.89, 19.39, and 637.3 J/g, as compared to 11.54, 18.22, and 522.7 J/g, respectively. Based on the literature, the high temperature can result in enhanced intermolecular reorientation of the chains to graft through MAH [13]. FTIR analysis also detected the depletion of amorphous regions to result in a strange increase of C-O-C bonds, which causes ease of mobility of C-O-C bonds to reorient into crystalline regions. The increase of crystalline enthalpies in DSC at high temperature, that resulted after degradation of the amorphous regions detected in FTIR, proved to be sufficient to increase the tensile strength in ANOVA main effects plots (Figure 2).

### 4.4. Measurement of Interlocking and Chemical Grafting

Thermogravimetric analysis (TGA) was used to validate the FTIR and DSC results regarding physical interlocking. The analysis also aimed to analyze the thermal stability to the degradation after soil biodegradation and moisture-based degradation. The analysis is provided in Figure 10 and Table 7.

Both non-printed and printed blends presented visible signs of excessive physical interlocking of PP in PLA (Figure 10). All thermographs showed a second step of PP at temperatures greater than 400 °C. As PP is 7.5% in the blend, the second step occurred at about less than 7.5% in both non-printed and printed blends. The occurrence of the second step at less than the added composition of PP confirms the limited chemical grafting and excessive physical interlocking [13]. The chemical grafting was probably formed between CH_3_ groups of PLA and C-H and C=O groups of MAH, as noted in FTIR analysis (Figure 8) [13]. The physical interlocking was confirmed by the bimodal melt crystallization peak in DSC analysis for non-printed and printed blends (Figure 9).

The water absorption samples showed a far lower onset temperature for the high-temperature combination (T_ONSET_ = 329.4 °C) and the low-temperature combination (T_ONSET_ = 335.3 °C) as compared to neat PLA (T_ONSET_ = 351 °C). The reason is the probable moisture degradation of the PLA amorphous regions in the blend, as presented in FTIR analysis (Figure 8). However, the physical interlocked PP, observed as the second step in TGA graphs (Figure 10), maintained the structural (mechanical) stability. Therefore, the overwhelming physical interlocking achieved better moisture-based stability for the blend with good mechanical properties.

## 5. Conclusions

This research presented the concept of partial chemical blending with excessive physical interlocking for fused filament fabrication (FFF). The research also presented an eco-friendly material with the minimum composition of non-biodegradable constituents ever reported in the literature, capable of providing better stability to in-process thermal variables and moisture-based degradation. In this regard, a ternary blend (PLA/PE-g-MAH/PP) was prepared with a minimum composition of PP (7.5%) and PE-g-MAH (0.5%). The research included various processing steps to overcome the unwanted degradation of the novel blend, for example, melt blending in a single-screw extruder and a specially modified 3D pellet printer with improved thermal insulation and cooling features. The novel blend was tested for in-process thermal and post printing moisture-based degradation. The in-process thermal degradation was analyzed with respect to the bed and printing temperatures. A statistical design of experiment was designed to statistically evaluate the effects of moisture-based degradation. FTIR, DSC, and TGA were used to analyze and validate the partial blending, and the reasons for stability.

The in-process thermal variables during 3D printing for the non-treated blend significantly affected the overall tensile properties, i.e., 32.5 MPa to 44.9 MPa.Different types of thermochemical characterizations proved the partial blending with minimum grafting and excessive physical interlocking. For example, FTIR showed that the distinct fourth peak of C-H groups associated with PP in the blend’s spectrum. DSC revealed the phase separation in the melt crystallization thermal profile of PP in the PLA thermograph. TGA showed a distinct two-step degradation profile, with PP as the second step (6.3% to 6.7%).The novel blend showed statistically high stability against moisture degradation.After being treated with 45 days of water absorption, the ANOVA provided the printing temperature as a significant variable, followed by moisture treatment. However, instead of a decrease, the tensile strength increased after water absorption treatment.The FTIR of moisture-degraded samples revealed the scission of chemical chains at the C-O-C bond. This chemical degradation was ound in DSC in the form of ΔH_M_ and ΔH_D_, and also in TGA as a decrease of onset temperatures.The study did not cover the hydrolytic degradation as a function of series of time. Instead, the moisture-based degradation was reported for a single duration of 45 days. Therefore, the complete potential of the blend is proposed for future research.

## Figures and Tables

**Figure 1 polymers-14-01527-f001:**
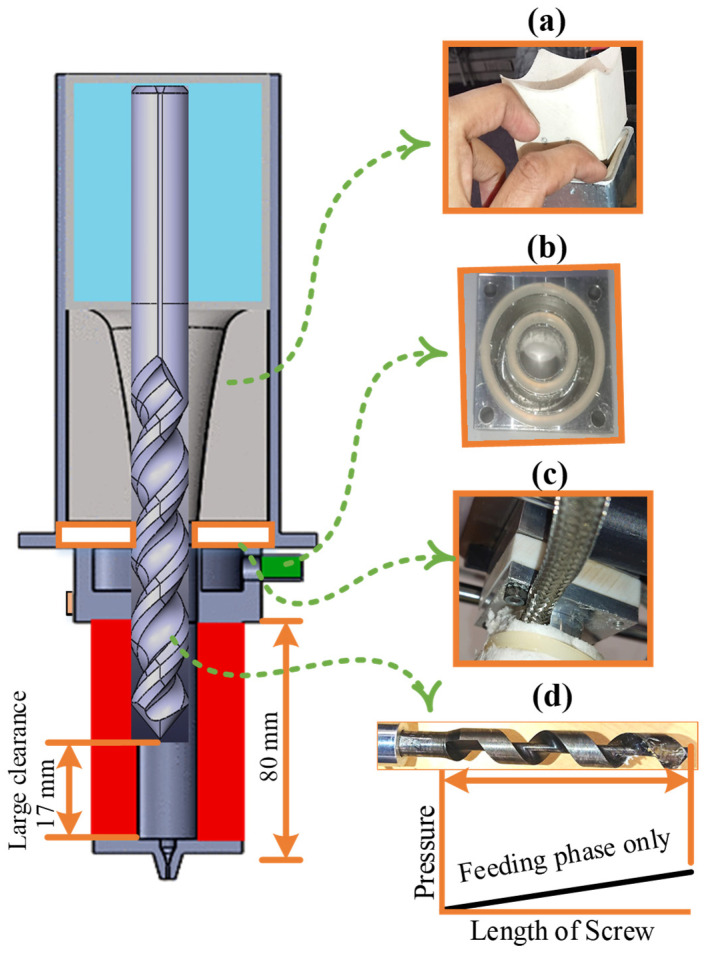
In-house-built 3D pellet printer [34] with modifications: (**a**) SLS 3D-printed cone, (**b**) large milled slot for liquid cooling, (**c**) Teflon plate for insulation, and (**d**) single profile extruder screw.

**Figure 2 polymers-14-01527-f002:**
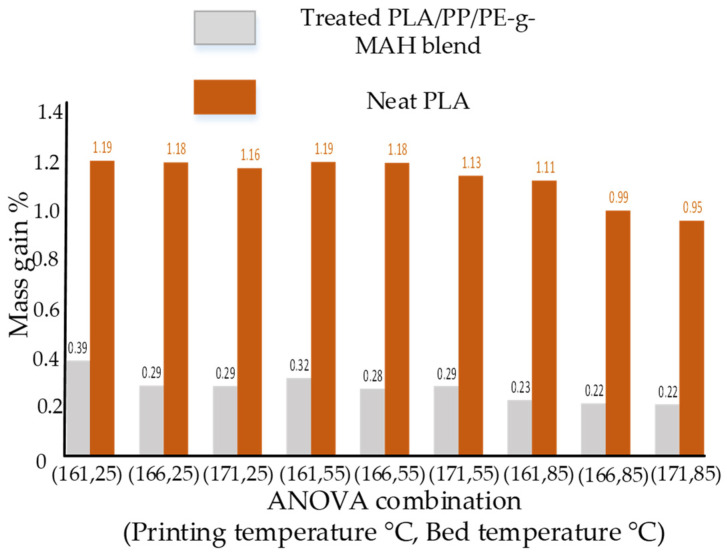
Mass gain percentage of water absorbed samples.

**Figure 3 polymers-14-01527-f003:**
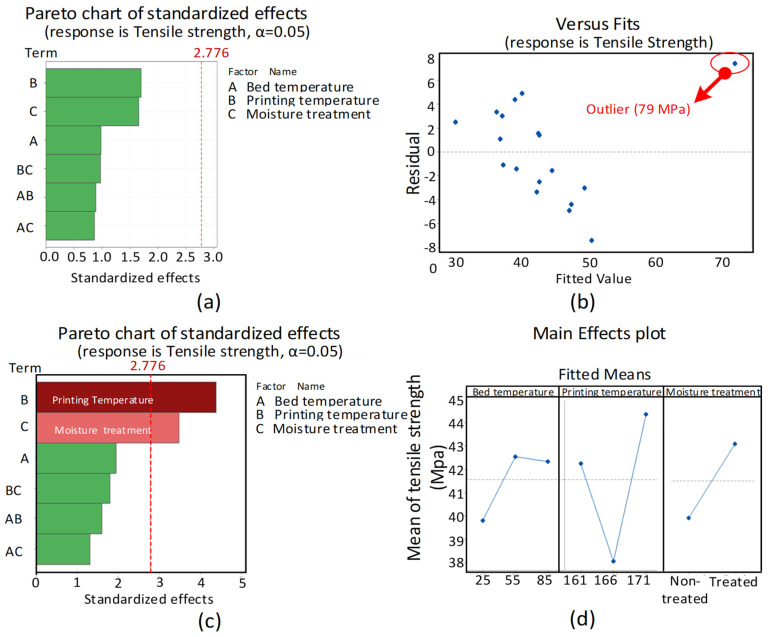
Results for water absorption: (**a**) pareto chart for first trial, (**b**) versus fits plot for first trial with outlier, (**c**) pareto chart for corrected trial, and (**d**) main effect plots for corrected trial.

**Figure 4 polymers-14-01527-f004:**
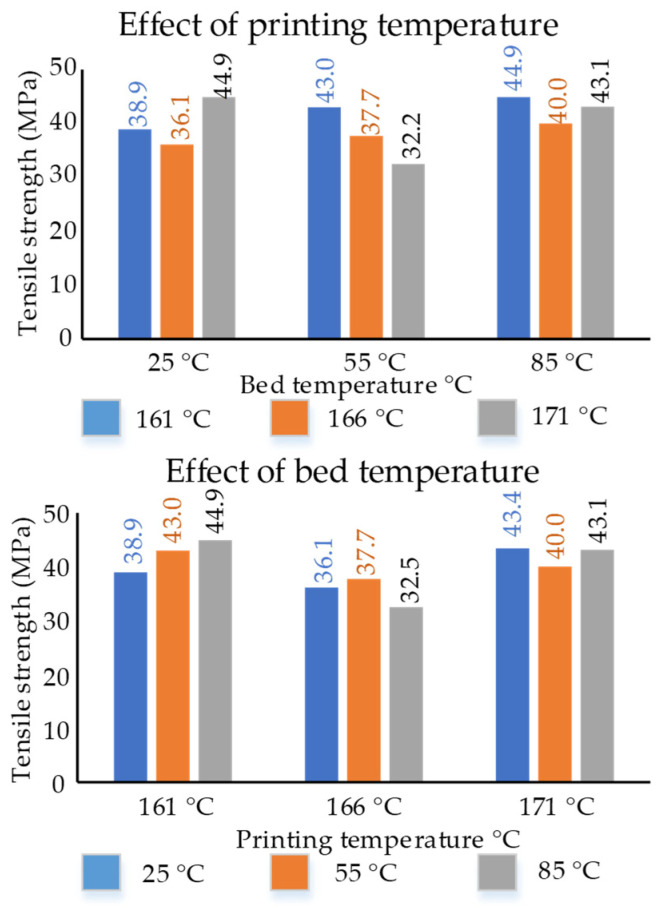
Effects of in-process thermal variables on tensile strength of ASTM D638 dog bones.

**Figure 5 polymers-14-01527-f005:**
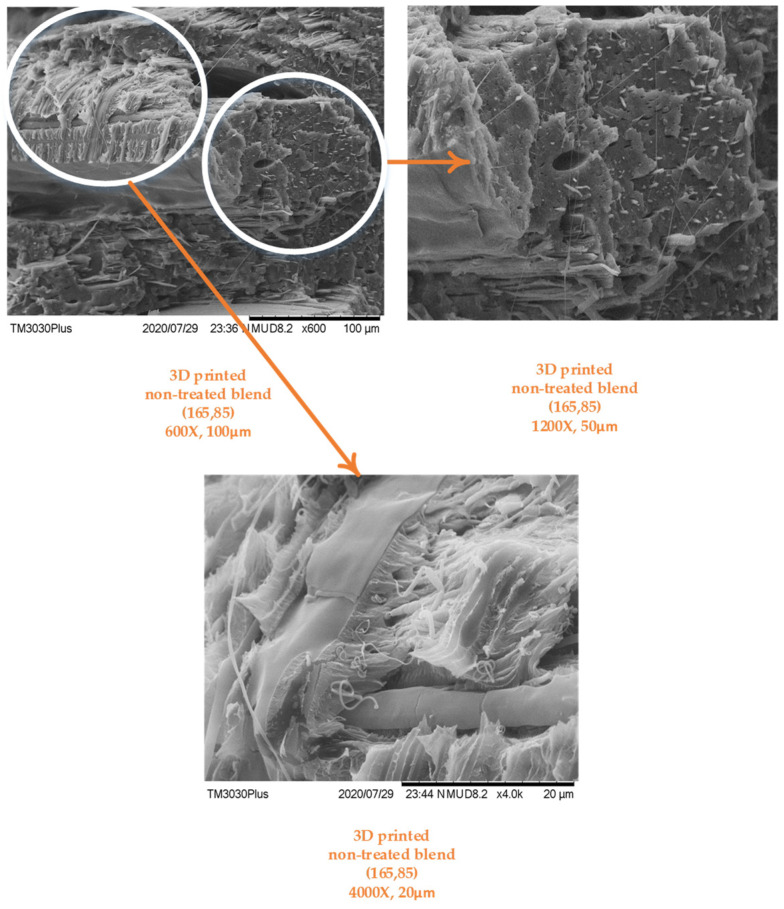
SEM analysis for melt blending in 3D-printed samples.

**Figure 6 polymers-14-01527-f006:**
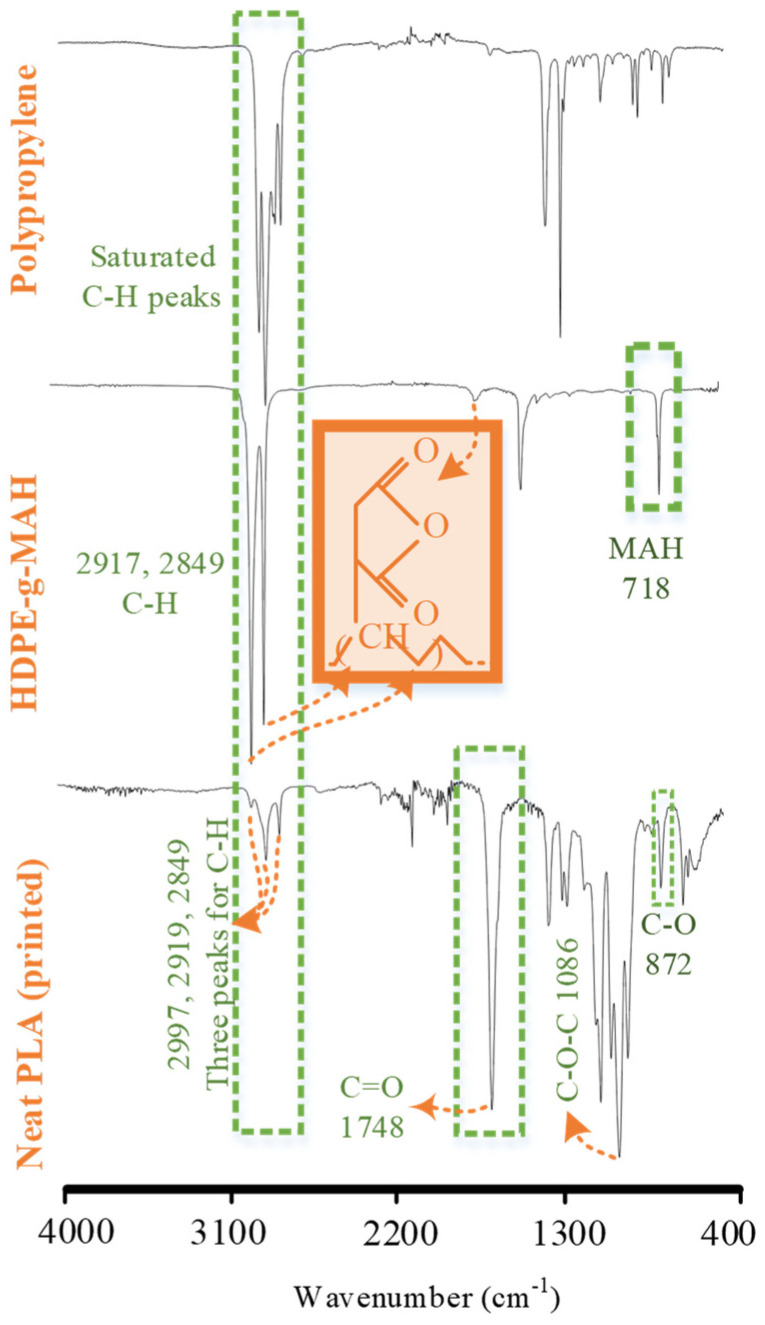
FTIR analysis of neat polymers.

**Figure 7 polymers-14-01527-f007:**
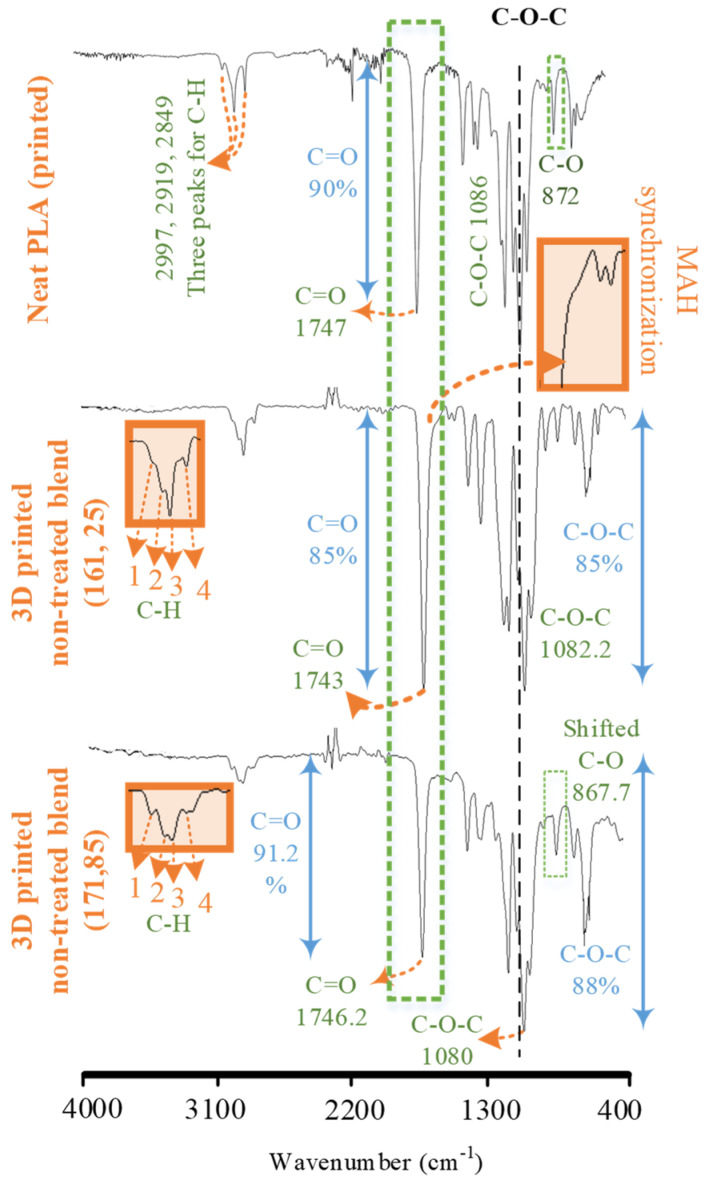
FTIR analysis of melt blending and 3D printing.

**Figure 8 polymers-14-01527-f008:**
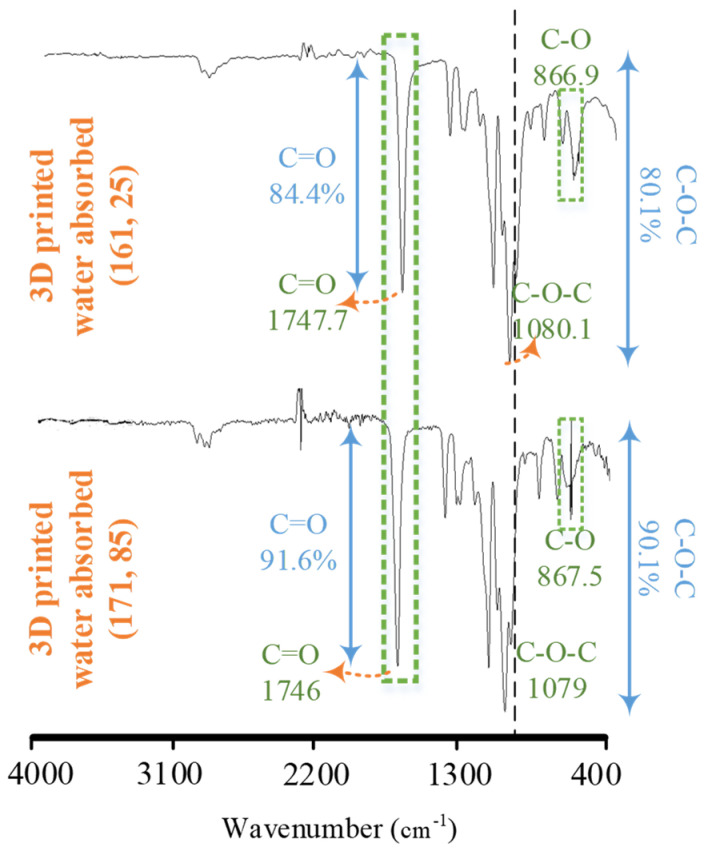
FTIR analysis for moisture degradation at the lowest and highest temperature combinations.

**Figure 9 polymers-14-01527-f009:**
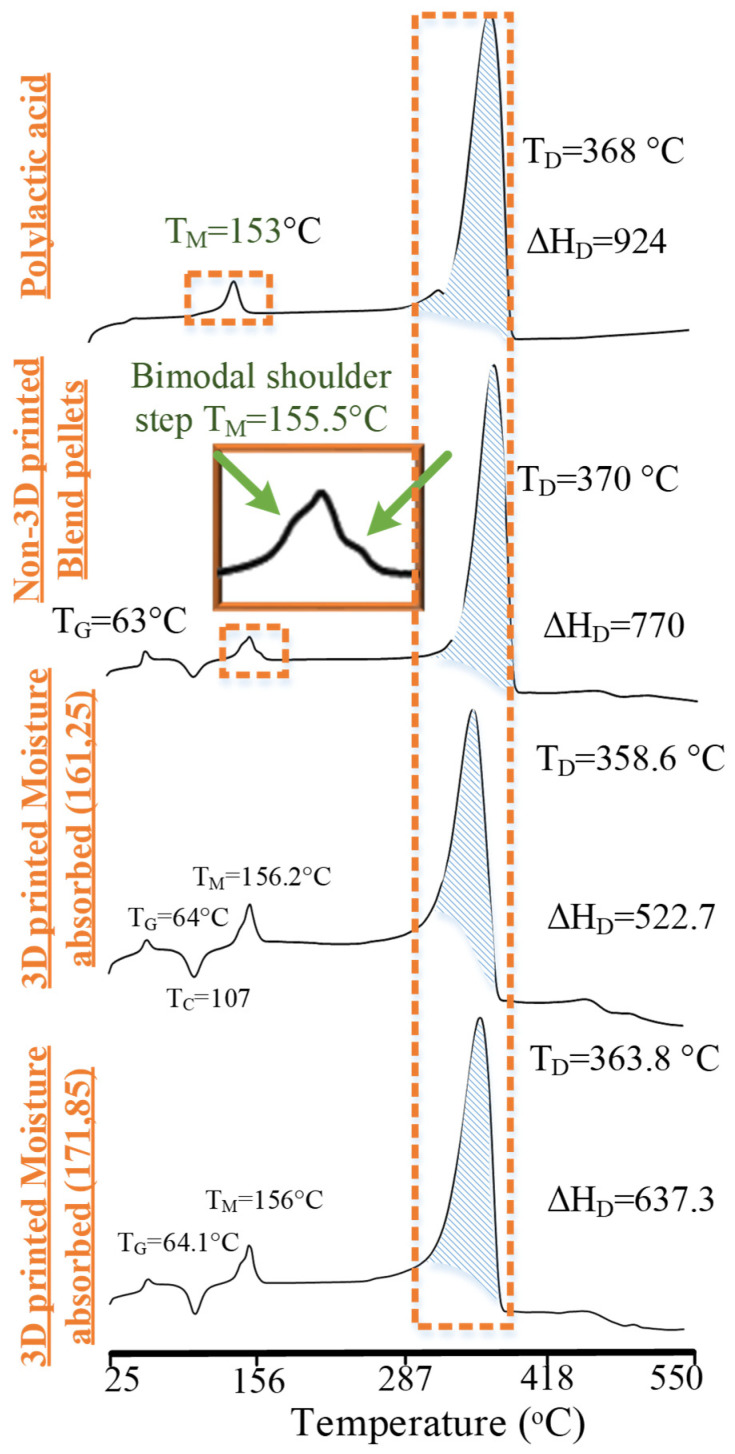
DSC analysis for the non-printed pellets and the 3D-printed blend.

**Figure 10 polymers-14-01527-f010:**
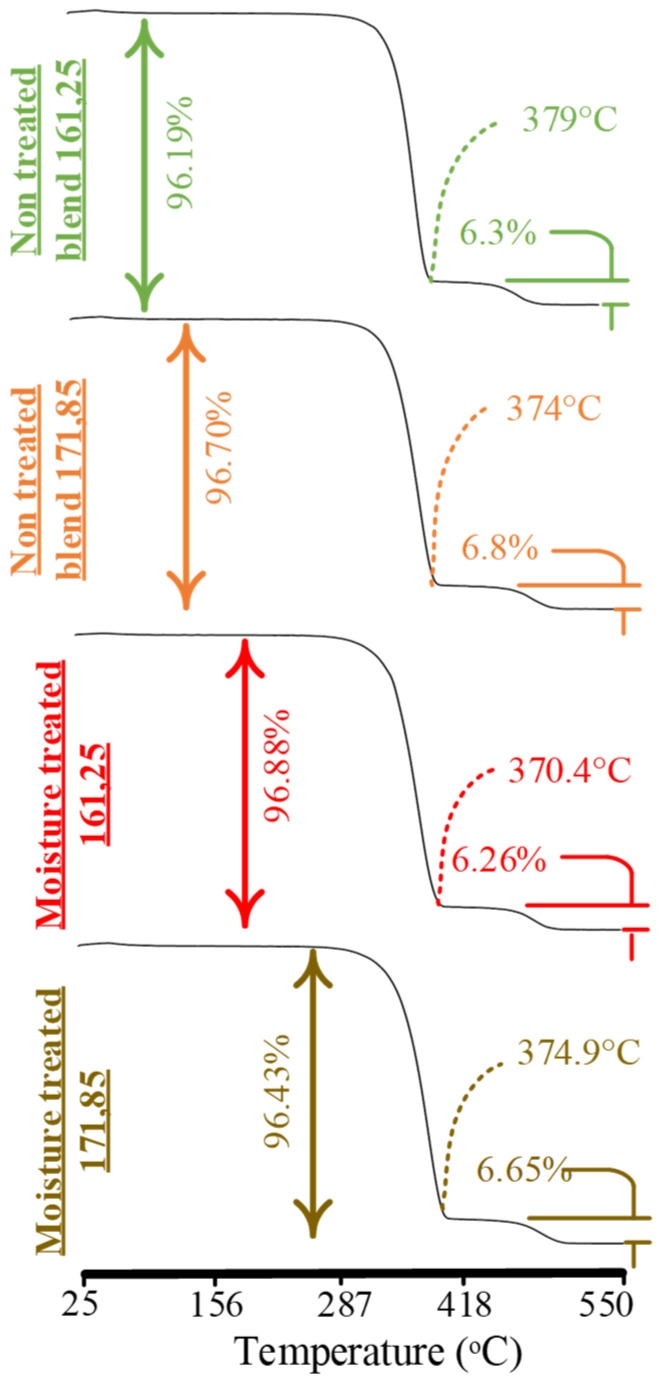
TGA analysis for soil biodegradation and moisture-based (water-absorbed) degradation.

**Table 2 polymers-14-01527-t002:** Compositions of blends prepared in the single-screw extruder.

Blend	PLA	PP	HDPE-g-MAH
**1**	75	20	5
**2**	92	7.5	0.5

**Table 3 polymers-14-01527-t003:** Parameters for 3D printing (FFF).

Parameter	Set Value
Layer thickness	0.2 mm [35]
Extrusion width	0.3 mm
Multiplier	5
Nozzle diameter	0.4 mm [12]
Printing speed	15 m/min
Bed temperature	25 °C, 55 °C, 85 °C
Printing temperature	161 °C, 166 °C, 171 °C
Infill density	100% [35]
Infill pattern	45°/−45° [35]

**Table 4 polymers-14-01527-t004:** General full factorial design of experiment (DoE) for water absorption analysis.

Factor (Parameter)	Level 1	Level 2	Level 3
Bed temperature (°C)	25	55	85
Printing temperature (°C)	161	166	171
Moisture absorption (Days)	0	45	

**Table 5 polymers-14-01527-t005:** DoE for second ANOVA of water absorbed samples.

RunOrder	PtType	Blocks	Bed Temperature	Printing Temperature	Moisture Treatment	Tensile Strength
**1**	1	1	25	171	Treated	43.00672
**2**	1	1	55	171	Non-treated	40.01403
**3**	1	1	55	166	Non-treated	37.71559
**4**	1	1	25	166	Non-treated	36.12446
**5**	1	1	85	166	Treated	40.15167
**6**	1	1	85	171	Non-treated	43.10712
**7**	1	1	55	171	Treated	46.4119
**8**	1	1	25	161	Treated	39.565995
**9**	1	1	25	171	Non-treated	43.37669
**10**	1	1	55	166	Treated	44.06397
**11**	1	1	25	166	Treated	37.79949
**12**	1	1	85	161	Treated	43
**13**	1	1	55	161	Treated	44.06397
**14**	1	1	55	161	Non-treated	42.99
**15**	1	1	85	171	Treated	50.3
**16**	1	1	85	161	Non-treated	44.9
**17**	1	1	25	161	Non-treated	38.92701
**18**	1	1	85	166	Non-treated	32.49289

**Table 6 polymers-14-01527-t006:** FTIR analysis (“WN” stands for wave number, cm^−1^, and “I” stands for intensity).

Material	Saturated C-H		C=O		CH_2_ and CH_3_ Bending Vibrations		Aromatic Styrene Ring		C-O-H		C-O-C		C-O		C-H	
	WN	I	WN	I	WN	I	WN	I	WN	I	WN	I	WN	I	WN	I
Neat PLA	2997 2919 2849		1747	90					1185		1086		872		729 755	
Water absorption blend (161, 25)	2993 2950 2919 2873 2838	97.7 97.3 97.6 98.1 98.2	1747.7	84.4	1451.2	93.8			1182.3	84.7	1080.1	80.1	866.9	93.5	675.6 750.3	91.4 93.3
Water absorption blend (171, 85)	2994 2949.8 2919 2849	98.7 98.6 98.6 99.1	1746	91.6	1451.2	96.3			1180.6	91.5	1079	90.1	867.5	96.9	664 754.5	96.4 96.9

**Table 7 polymers-14-01527-t007:** TGA analysis for the PLA/PP/PE-g-MAH blend.

Material	Onset Temperature °C	End Temperature °C	First Mass Loss % (M_A_)	Second Mass Loss % (M_B_)	Total Mass Loss in Two Steps % (M_T_ = M_A_ + M_B_)	Mass Remained (100%-M_T_) %
PLA	351	393	96.73	0	96.73	3.27
Pellets	348	380	90.24	6.7	96.97	3.03
Moisture (161 °C, 25 °C)	329.4	370.4	90.62	6.26	96.88	3.12
Moisture (171 °C, 85 °C)	335.3	374.9	89.78	6.65	96.43	3.57

## Data Availability

Not applicable.

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
