# Peer review of "Partial Biodegradable Blend for Fused Filament Fabrication: In-Process Thermal and Post-Printing Moisture Resistance"

_polymers, 2022, doi:10.3390/polym14081527_

Round 1

Reviewer 1 Report

The Authors revised their manuscript following the suggestions of my first review round, thus in my opinion the manuscript is suitable for publication on Polymers.

Author Response

We are grateful to the respected reviewer to approve our document for publication.

Reviewer 2 Report

The article has been significantly improved. It is still unclear, why the hydrolytic degradation, which is the basis of the article - as the title indicates, has not been investigated as a function of time. This needs to be completed. 

Author Response

The reviewer’s comments are addressed in version 3. We are thankful to the respected reviewer to point out the possible changes to make the quality better. In this regard, we have emended our document and now the document is based upon two degradation phenomenon, i.e., in-process thermal variables and moisture-based degradation for 45 days. The newly added section is “2.5. In-process (during 3D printing) thermal testing” covered in lines 263 to 278. The corresponding results are presented in section “3.3. Effects of bed and printing temperatures” from lines 352 to 363.

The time-based hydrolytic degradation is proposed for future research in Conclusions (lines 559 to 562).

Reviewer 3 Report

The authors had been asked to merger two parts. But, they did not. 

Author Response

The authors had been asked to merger two parts. But, they did not. 

We are grateful to the Reviewer for the valuable comments. We have considered the comment seriously and modified as much as possible within available results, funding and consensus of research group.

The reviewer showed reservations on the novelty of reported results as compared to the previous publication by same group of authors. The reviewer also highlights the non-suitability of article in previous version for high impact factor “Polymers journal”.

In this regard, we have now added a new analysis in “methodology” and the corresponding results in “results”. New section of “2.5. In-process (during 3D printing) thermal testing” is added in lines 263 to 278. The corresponding results is now covered in “3.3. Effects of bed and printing temperatures” from lines 352 to 363. The newly added sections are not being reported ever before.

Furthermore, the version 2 was also added with SEM analysis to compliance with the suggestion by the reviewer. SEM analysis is particularly performed to fulfill the requirements of reviewer. In current form, the new version 3 is now significantly improved by addition of above-mentioned sections.  

We have also modified the title to “Partial biodegradable blend for fused filament fabrication: In-process thermal and post-printing moisture resistance”.  The modified title now covers two aspects of in-process thermal variables and moisture treatment for 45 days instead of just considering hydrolytic degradation. Both of these aspects are covered in methodology and results.

The time-based hydrolytic degradation is also proposed for future research in Conclusions (lines 559 to 562).

Round 2

Reviewer 2 Report

Accept in present form.

Reviewer 3 Report

The merged manuscript is recommended for publication in its merged form.

This manuscript is a resubmission of an earlier submission. The following is a list of the peer review reports and author responses from that submission.

Round 1

Reviewer 1 Report

The topic of biodegradable blends for FDM are well-studied, e.g., see the references. The hydrolytic degradation is very known too. The submitted manuscript and the published papers are very similar. The present manuscript does not include any novel results or discussion and does not deserve a new paper in a high impact journal such as Polymers. Perhaps the author should merger two parts. In conclusion, the innovation of the manuscript is scarce.

Reviewer 2 Report

The manuscript is part of a wide research carried out by these Authors on the design, preparation and characterization of biomaterials for FFF. The manuscript is well written and presents (in the introduction) a quite complete state of the art (probably some statements could be strengthened by adding specific references). I would suggest some changes highlighted in the attached document.

Reviewer 3 Report

The article “Part 2: Partial biodegradable blend with high stability against  hydrolytic degradation for fused filament fabrication” covers research on PLA/PP blends with MA compatibilizer. Such systems are not new and are known from the literature. The article has a number of flaws and therefore requires extensive revision. In addition, only two sample compositions were tested.

Specific comments:

Abstract is too general. It should be remodeled and focused more on the results than techniques’ description.

Line 39 – “various reasons” have to be specified.

Line 53 – the abbreviation PLA should be placed at the first appearing of the full name in the manuscript.

Line 96 - HDPE-g-MA – full name is lacking.

MA and MAH abbreviations have to be uniformed.

Line 104 – “Scanning electron microscopy (SEM) analysis.” – the methodology, results and discussion were not discussed at all.

Lines 111-112 – “The high MFI grade is used to 111 ensure proper melty blending while considering the intentional overwhelming physical 112 interlocking [12].” – it better suited to the discussion.

Paragraph: 2.2. Melt Blending – provide blending parameters, temperature and pressure.

Table 1 - small variety of samples (2 sample composition).

Explain why specifically such sample compositions were used for the research.

2.5. Tensile testing – describe specimen parameters (shape/dimensions) and procedure of their manufacturing.

Experimental part - there are no countries and cities of production of apparatus used in the work.

Results is too poor. There is only one section: 3.1. Water absorption (hydrolytic degradation). The results obtained by other methods have not been described.

In fact, the hydrolytic degradation has not been measured. It is done by observing water uptake as a function of immersion time.

The authors may consider cutting out the DSC thermograms, showing the glass transition temperature and melting temperature. Now, the entire temperature spectrum is shown on one thermogram, therefore the results for Tg and Tm are hard to analyze.

The discussion of the DSC results should be based on the thermal properties of PP and PLA. Thermodynamic aspect of mixing should be discussed too.